# The Role of Exercise on Cardiometabolic Profile and Body Composition in Youth with Type 1 Diabetes

**DOI:** 10.3390/children9121840

**Published:** 2022-11-28

**Authors:** Maria Kaza, Charalampos Tsentidis, Elpis Vlachopapadopoulou, Spyridon Karanasios, Irine-Ikbale Sakou, George Mastorakos, Kyriaki Karavanaki

**Affiliations:** 1Diabetes and Metabolism Unit, 2nd Department of Pediatrics, National and Kapodistrian University of Athens, “P&A Kyriakou” Children’s Hospital, 11527 Athens, Greece; 2Department of Endocrinology, Metabolism & Diabetes Mellitus, General Hospital of Nikaia-Piraeus “Agios Panteleimon”, 18454 Piraeus, Greece; 3Department of Endocrinology Growth and Development, “P&A Kyriakou” Children’s Hospital, 11527 Athens, Greece; 4Endocrine Unit, “Aretaieion” Hospital, National and Kapodistrian University of Athens, 11528 Athens, Greece

**Keywords:** adolescence, T1D, pediatric, leptin, adiponectin, IL6, hs-CRP, physical activity, pedometers, fat mass, muscle mass

## Abstract

Exercise has a direct positive effect on glycemic control by promoting insulin secretion from β-pancreatic islet-cells and by increasing skeletal muscle glucose uptake. The reduction in daily insulin needs and the optimization of glycemic control improves the patient’s quality of life, self-esteem, mental wellness, as well as diabetes-related mobility and mortality. The aim of this study was to investigate the effect of physical activity in children and adolescents with type-1 diabetes (T1D) on diabetic control, cardiovascular, and biochemical profiles; hs-CRP; IL6; leptin; and adiponectin levels of the population under study. This is a prospective cross-sectional study that involved 80 participants (36 boys and 44 girls) with T1D, who were aged 6–21 years and who attended the Diabetes and Metabolism Clinic of the 2nd Pediatric Department, University of Athens, “P & A Kyriakou” Children’s Hospital of Athens. Twenty (25%) children were above the 75th percentile regarding total levels of physical activity, while 40 (50%) and 20 (25%) were between the 25th and 75th percentile, as well as below the 25th percentile, respectively. In the group with an intermediate level of exercise, physical activity was negatively associated with the participant’s family situation (traditional, single parent, grandparent, with others, or by himself/herself) (*p* = 0.013), ferritin (*p* = 0.031), lipoprotein(a) [Lp(a)] (*p* = 0.016), and squared leptin levels (*p* = 0.040). Whereas in the groups with extreme vs. no exercise there was a negative association with the number of daily glucose measurements (*p* = 0.047). However, in the group with non-vigorous exercise, physical activity was positively associated with high density lipoprotein-c (HDL-c) levels (*p* = 0.048). The findings of this study are indicative of the beneficial role of exercise on children and adolescents with T1D, which is achieved by primarily improving their cardiometabolic profile through the amelioration of lipid profile [HDL-c, Lp(a)] and leptin levels, as well as by reducing chronic systemic inflammatory response (ferritin) and ultimately decreasing the overall diabetes morbidity.

## 1. Introduction

Individuals with type 1 diabetes mellitus (T1D) have at least double the risk of developing cardiovascular disease when compared with the non-diabetic population [1]. Exercise promotes insulin secretion from β-pancreatic islet-cells and increases skeletal muscle glucose uptake [2]. The reduction in daily insulin needs and the optimization of glycemic control improves the patient’s quality of life, self-esteem, mental wellness, and diabetes-related morbidity and mortality [2,3,4]. Studies on the physical activity of youth with T1D are limited [5,6,7,8,9,10,11,12,13,14,15,16,17,18,19,20,21,22,23,24,25,26]. However, overall they conclude that regular physical activity is responsible for improved cardiorespiratory capacity, body composition, bone health, insulin sensitivity, and psychosocial well-being [27].

The positive effect of exercise on childhood and adult obesity, based on the results of research studies and meta-analyses, is indisputable [28,29,30]. Although it is recommended that all children and adolescents (5–18 years old) should exhibit an hour of moderate to vigorous physical activity every day, 1 in 3 adolescents do not meet this target [27]. It is estimated that about 50% of adult patients with T1D are either obese or overweight and the possibility that they will undergo bariatric surgery in the future will be commensurate with the general population or patients with Type 2 Diabetes (T2D) [31]. As such, healthy diet and physical activity are the major key regulators of T1D management.

Lipoprotein(a) [Lp(a)] is a highly atherogenic and thrombogenic low-density lipoprotein. It consists of apolipoprotein(a) bound to apolipoprotein B by a disulfide bridge [32]. High density lipoprotein-c (HDL-c) demonstrates important functional atheroprotective properties. These include a major involvement in macrophage cholesterol efflux and reverse cholesterol transport, which is a protective role against the arterial wall inflammatory process. Moreover, this is also the case with the mitigation of oxidative modifications of low density lipoprotein-c (LDL-c), which is the major component of atheromatic plaque [33,34]. Previous studies have reported that regular exercise can improve patients’ lipid profile, body composition, fitness, and glycemic control. It can also improve their general health and wellbeing [35].

Adipokines are a large family of hormones, secreted by adipose tissue cells, that hold a key role in weight regulation. Leptin (in the Greek language leptos means thin), the first adipokine to be discovered in the 90s was produced mainly by fat cells. As an adipokine, leptin is a primary regulator of energy expenditure, weight maintenance, and a prognostic indicator of cardiovascular risk [36]. As a cytokine, leptin has pro-inflammatory properties and regulates the secretion of inflammatory cytokines [37]. Decreased leptin levels are commonly associated with reduced cellular immunity [38]. Given the effect of physical exercise on body composition, there are very limited previous studies associating leptin levels with the degree of physical exercise in children and adolescents with T1D [39].

This study assessed the effect of physical activity (divided into groups according to the quantity of physical activity) on a cardiometabolic profile, inflammatory cytokines, adipokines [interleukin 6 (IL6), leptin, and adiponectin], and also infection indices [high-sensitivity C-Reactive-Protein (hs-CRP)] in youth with T1D.

## 2. Materials and Methods

### 2.1. Subjects

Thirty-six (36) boys and forty-four (44) girls with T1D, aged 6–21 years were included in this cross-sectional study. All participants attended the Diabetes and Metabolism Clinic of the 2nd University Pediatric Department of the “P & A Kyriakou” Children’s Hospital of Athens [40]. The criteria for inclusion in the study were: age range between 6–21 years; a duration of T1D at least at 12 months since diagnosis and follow-up by the abovementioned Clinic, between the years 2019 and 2020. The criteria for exclusion from this study were: children with non-T1D; a shorter than 12 months duration of T1D; children with mobile-disabilities or other chronic untreated conditions; children with acute febrile illness at the time of this study; and finally, children on statins or anticontraceptive pills.

This study assessed the effect of physical activity (divided into 3 groups according to quantity of physical activity) on the participants’ metabolic and biochemical profile, body composition, and the levels of adipokines (leptin and adiponectin), IL6, and hs-CRP. The level of physical activity was evaluated with pedometers for 7 days in total (via an Omron Walking Style IV).

### 2.2. Protocol

All participants (>18 years old) or their legal guardians if they were minors (<18 years old) filled and signed an informed consent form. The details of their medical documents, as well as the data of their most recent laboratory blood results, were acquired from their medical records (i.e., their most recent HbA1c%, biochemical, hematological, and inflammatory blood results). The mean HbA1c% was calculated as the mean of three consecutive measurements from the past 12 months. During the participants’ visit at the clinic, mean diastolic and systolic blood pressure (BP) were measured as the mean of two measurements. Moreover, height (cm) was measured with a horizontal stadiometer by an experienced practitioner (M.K) and the body mass index (BMI) was calculated with the Quetelet formula [BMI (kg/m^2^) = mass (kg)/height (m)^2^]. The standard deviation score (SDS) height, as well as the z-score BMI were calculated using national normative data [41]. The participant’s pubertal status was evaluated and categorized according to a visual Marshall and Tanner scale by an experienced physician (M.K) [42,43]. Their body composition was assessed with the Bioelectrical Impedance B.I.A (Tanita, Body Composition Analyzer, Type BC-418 MA) [44,45,46,47]. The level of patients’ exercise was assessed using pedometers (Omron Walking Style IV) measuring total number of steps per week. All participants answered an anonymous questionnaire. Blood samples were obtained after an 8 h overnight fast in order to assess the levels of leptin, adiponectin, IL6, and hs-CRP. Immediately after collection the samples were centrifuged and stored in −70°. Subjects’ laboratory parameters were derived from their latest appointment at the clinic (if this occurred during the preceding week of pedometers’ placement), and blood samples for adipokines, IL-6, and hs-CRP were obtained. If there was no appropriate latest appointment, then further blood samples were taken and processed within the following week.

### 2.3. Assays

Serum blood samples were obtained from the children after an 8 h overnight fast and were immediately centrifuged and stored in −70 °C [40]. Following slow thawing: (i) Serum leptin samples were processed with a leptin ELISA kit (Biovendor 62100 Brno Czech Republic) with an intra assay variation CV% of 4.2%, inter assay variation CV% 6.7%, and a detection limit 0.2 ng/mL; (ii) adiponectin serum samples were processed with an ELISA kit (Invitrogen, Campus Vienna Biocenter 2, 1030,Vienna, Austria), with an intra assay variation CV% of 4.2%, an inter assay variation CV% of 3.1%, and a detection limit: <0.012 ng/mL; (iii) the hs-CRP serum samples were obtained via the use of a hs-CRP ELISA kit (HyCult Biotech, Frontstraat 2A, 5405 PB Uden, Netherlands), with intra assay variation CV% 4.1%, with an inter assay variation CV% of 6.3%, and a detection limit < 0.4 ng/mL; and (iv) for measuring the IL-6 levels an IL6 ELISA kit (Origene Tech., 9620 Medical Center Drive, 200, Rockville, MA, USA) was used with intra assay variation CV% 6.2%, inter assay variation CV% 7.2%, and a detection limit < 0.3 pg/mL [40].

### 2.4. Ethics

All children participated voluntarily in this study and signed an informed consent if older than 18 years of age or an ascent if they were younger. For participants that were under eighteen years old at the time this study took place, a written consent was requested from their guardians. Participants were able to withdraw their consent at any stage of this study. The confidentiality of medical records was maintained. No ethical concerns were risen due to the observational nature of this study. The study was approved by the Ethics Committee of the “P&A Kyriakou” Children’s Hospital, Athens, Greece, protocol number: 12919, date of approval: 24 July 2019 [40].

### 2.5. Statistical Analysis

Data handling and statistical analyses were conducted using SPSS (version 23) and STATA SE for Windows v11.2, (StataCorp, Austin, Texas, TX, USA, 2012). Furthermore, the data were presented as actual numbers (i.e., percentage proportions) for categorical variables, mean ± SD, median, and interquartile ranges for the numerical variables. All numerical variables were analyzed both graphically and statistically using the Shapiro–Wilk criterion. While most variables were found to adequately approximate normal distribution, mathematical transformation was necessary in the four variables of interest. Leptin was square root transformed, adiponectin and CRP were log transformed, and IL-6 was inversed [40].

Univariate linear regression models were used in order to study associations among leptin, adiponectin, CRP, and IL-6 with demographic, clinical, and biochemical variables. The comparison between the two extreme exercise groups was performed using the ANOVA (analysis of variance) method with Bonferroni correction, or with Fisher’s exact test for numeric or categorical variables, respectively.

Furthermore, a *p*-value of < 0.05 was considered significant.

## 3. Results

This study’s demographic, clinical, and biochemical information were adapted from Kaza M. et al. [40]. This study included eighty children with T1D [36 (45%) boys, age: mean ± SD, 14.89 ± 3.44 years]. Mean disease duration was 5.8 ± 4.03 years, mean annual HbA1c: 8.0 ± 1.41. Twenty (25%) children were above the 75th percentile regarding total levels of physical activity (high level of exercise group), while 40 (50%) were between the 25th and 75th percentile (intermediate level of exercise group), and 20 (25%) were below the 25th percentile (low exercise group), respectively.

Within the intermediate level of the exercise group of patients (Table 1), the level of exercise was negatively correlated to the participants’ family situation (traditional, single parent, grandparent, with others, or by himself/herself) (beta = −0.2785692 and *p* = 0.013); serum ferritin (beta = −0.0123 and *p* = 0.031); lipoprotein(a) [Lp(a)] (beta = −0.0054 and *p* = 0.016); and square leptin (beta = −0.1036 and *p* = 0.040). There were no significant correlations found in regard to muscle mass (beta = 0.00393 and *p* = 0.527) and fat mass % (beta= −1.6156 and *p* = 0.133).

Moreover, regarding the physical activity in the non-vigorous exercise group (<75% percentile group), the amount of exercise was positively associated with HDL-c (beta = 0.014 and *p* = 0.048) (Figure 1).

When studying the group of patients with extreme levels of exercise (low and high exercise groups), total number of steps per week were negatively associated with the number of daily glucose measurements (beta = −0.075 and *p* = 0.047), ferritin (beta = −0.011 and *p* = 0.023), and Lp(a) levels (beta = −0.004 and *p* = 0.035).

Moreover, when comparing the two extreme exercise groups (a. low exercise levels and b. high exercise levels) (Table 2), patients in the high exercise group had fewer daily glucose measurements (*p* = 0.047), lower ferritin levels (*p* = 0.022), lower log Lp(a) (*p* = 0.007), and lower SqR(Leptin) levels (*p* = 0.059). No significant correlations were found with muscle mass (*p* = 0.51) and fat mass % (*p* = 0.10). It is noteworthy that in the high exercise patient’s group, family situation and diabetes duration were only marginally not statistically significantly, which is different from the low exercise group (Table 2).

When comparing the three exercise groups using ANOVA with the Bonferroni correction, hemoglobin levels were significantly related to the degree of physical activity, with the highest levels in the high exercise group (13.9 ±1.2 vs. 13.2 ± 1.4 vs. 14.0 ± 1.06 gr/dl, and *p* = 0.03). Moreover, in Figure 2a it is shown that serum leptin levels were lower in the high exercise group and were progressively increased inversely to the amount of exercise, with the highest levels in the low exercise group. Moreover, no significant difference in terms of adiponectin levels among the three exercise groups was found (Figure 2b).

## 4. Discussion

Studies investigating physical activity in children and adolescents with T1D are limited [5,6,7,8,9,10,11,12,13,14,15,16,17,18,19,20,21,22,23,24,25,26]. In addition, the majority of them conclude that physical activity has a positive effect on daily insulin requirements, HbA1c%, lipid profile, and the prevalence of impaired indices of diabetic microangiopathy [2,48,49,50].

This is a cross-sectional study incorporating objective means (pedometers) for evaluating the level of physical activity in children with T1D. To the best of our knowledge there is no similar completed study that is up-to-date; that assesses the effect of physical activity (subdivided into groups according to the amount of exercise) on cardiometabolic profile, body composition, adipokines, IL6, and hs-CRP levels [51]. Furthermore, this is one of the few studies to date where a bioelectric embedding scale was used to analyze the impact of physical activity on the body composition of the pediatric population with T1D [52,53,54].

The majority of studies on youth with T1D conclude that children and adolescents with T1D are less physically active than their healthy peers [55,56]. In the present study, 25% children and adolescents with T1D aged 6–21 years were significantly physically active, whereas 25% demonstrated levels of reduced or no physical activity.

According to the results of this study, the level of physical activity is strongly associated with improved lipid levels in children and adolescents with T1D. Of great interest, however, is the novel finding of the present study that there is negative linear correlation of physical activity with Lp(a) values. This finding is in agreement with the suggestion of Rigla et al. [57], who stated that regular exercise could indeed improve the cardiometabolic profile of T1D patients by lowering the levels of Lp(a).

Another finding of this study was that in the group of patients without vigorous exercise, a strong positive correlation was found between the level of physical activity and HDL-c levels. HDL-c demonstrates important functional atheroprotective properties, among others through the restriction of LDL-c oxidation [49,50]. Although the positive correlation between moderate to vigorous physical activity and HDL-c was reported in the review of Leclair et al. in regard to children and adolescents with T1D [58], this study’s finding that non vigorous exercise also has a positive effect on HDL-c levels is novel and has significant clinical implications for children and adolescents with T1D.

Leptin is an adipokine, produced by adipose tissue, which has been associated with weight maintenance, inflammation, cardiovascular risk, and reduced cellular immunity [52,53,54]. In this study, leptin was negatively correlated to the intensity of physical activity. This finding is in agreement with the Leclair et al. review in children and adolescents with T1D [58], which reports that six months of aerobic and muscle strength training was able to limit leptin levels, indicating the beneficial role of exercise in the cardiometabolic profile improvement of this population.

Serum ferritin is as an acute phase protein and an inflammatory marker, rising in a variety of acute and chronic inflammatory conditions, such as autoimmune disorders and acute infection. Ongoing research results suggest that elevated serum ferritin levels are a risk factor for coronary artery disease (CAD) [59]. Interestingly, this study reports a negative correlation between physical activity and ferritin levels, which suggests that regular intense exercise contributes to the reduction in chronic inflammation in children and adolescents with T1D. Another explanation of the latter negative correlation between exercise and serum ferritin levels is given by Bartfay et al. [59], who suggested that regular aerobic exercise may decrease iron stores in the healthy adult population [59].

Interestingly, hemoglobin was strongly related to the levels of physical activity as shown by this study’s results after applying an analysis of variance (ANOVA) of the three groups of exercise (low, intermediate, and high). This finding may be explained by the physiological alterations in plasma volume that may cause changes in hematocrit and hemoglobin in individuals that exercise regularly, such as athletes [60]. Moreover, our finding that the high exercise group presented increased hemoglobin levels could also explain lower ferritin levels in these patients.

According to this study’s results, the more physically active participants had reduced requirements for frequent daily glucose measurements. This could be possibly explained by the fact that physically active children tend to use CGMs (continuous glucose monitoring devices) in order to regularly follow their blood glucose levels before, during, and after exercise [61,62,63]. This is probably the reason that regular exercise was associated with fewer blood glucose measurements in our study. However, according to the study of Hansen et al., adults with T1D demonstrate overall limited compliance regarding self-monitoring of blood glucose [64].

Finally, this study’s participants’ family situation was correlated to measured levels of their engagement in physical activity. Participants living by themselves, with a single parent, or grandparents were less active than those living with both parents. To our knowledge, there are no previous studies on the association of family situation with the presence and the degree of physical exercise in children and adolescents with T1D. However, this finding is in agreement with the Wen LK study on older adults with T2D [65], in which it was reported that higher levels of patients’ perceived family support and self-efficacy were associated with higher levels of exercise [65].

Among the advantages of the present study was that it is among the very few studies with quantitative measurements of the level of exercise with the use of pedometers. Moreover, it is also one of the very few studies that has investigated the impact of physical activity on body composition using bioelectric embedding scales. Having said this, a disadvantage of our study was that it did not include a control group for the comparison of the biochemical and clinical parameters between T1D patients and their non-diabetic peers.

## 5. Conclusions

Overall, the findings of this study are indicative of the beneficial role of exercise on the cardiometabolic profile of children and adolescents with T1D. Moreover, the intensity and frequency of exercise ameliorates participants’ lipidemic profile [Lp(a) and HDL-c], which reduces the chronic systemic inflammatory response (based on serum ferritin levels) and ultimately, positively affects diabetes-related morbidity.

## Figures and Tables

**Figure 1 children-09-01840-f001:**
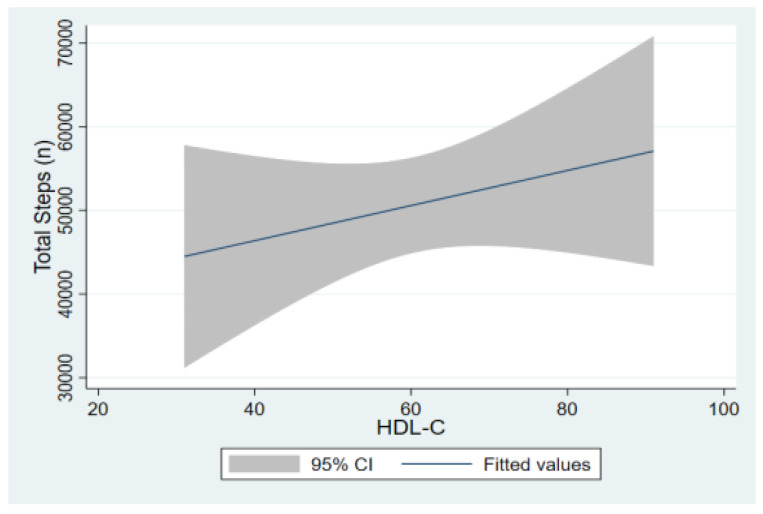
Association of the amount of physical exercise with the patients’ HDL levels in patients with non-vigorous exercise.

**Figure 2 children-09-01840-f002:**
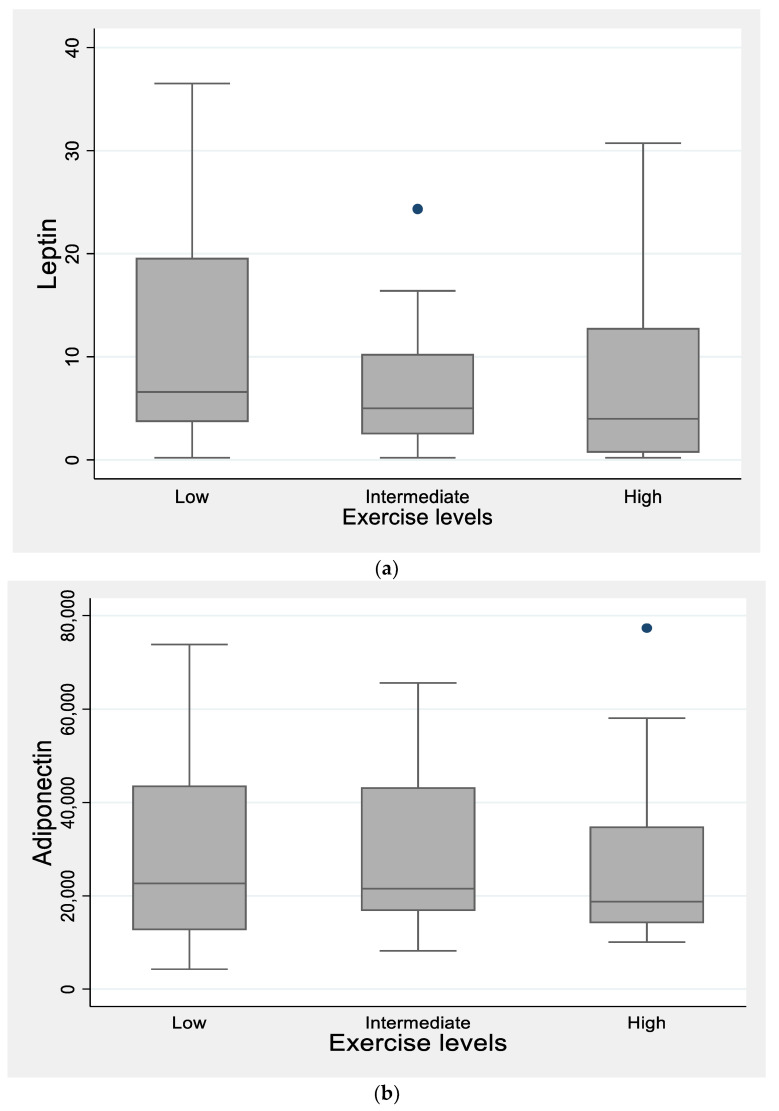
(**a**). Leptin levels were progressively increased inversely to the level of exercise, with the lowest levels in the high exercise group. (**b**). The three exercise groups were not significantly different in terms of adiponectin levels.

**Table 1 children-09-01840-t001:** Total steps 25–75% level of physical activity group (intermediate exercise level): correlations with basic demographic clinical and biochemical parameters with the use of univariant linear regression.

Px	Beta	(95% CI of Beta)	T-Statistic	*p*-Value
Demographic and anthropometric parameters				
Gender (female)	−0.101	(−0.420, 0.2185)	−0.63	0.531
Chronological age (years)	0.009	(−0.0365, 0.0564)	0.43	0.671
Tanner (II–V)	−0.0301	(−0.172, 0.1126)	−0.42	0.675
Height (cm)	0.0002	(−0.0068, 0.0073)	0.08	0.939
Weight (kg)	0.0003	(−0.01003, 0.0106)	0.06	0.952
Waist circumference (cm)	−0.0008	(−0.0161, 0.0144)	−0.11	0.912
Hip circumference (cm)	0.0116	(−0.0063, 0.02961)	1.29	0.202
Waist/hip ratio	−0.50512	(−1.303594, 0.2933)	−1.26	0.212
Waist/height ratio	−0.00003	(−0.03779, 0.0377)	−0.00	0.999
BMI (kg/m^2^)	−0.0009	(−0.0443, 0.04245)	−0.04	0.966
z-score BMI (SD)	0.001	(−0.1706, 0.1728)	0.01	0.990
Muscle mass (kg)	0.00393	(−0.0083, 0.0162)	0.64	0.527
Fat mass percentage (%)	−1.6156	(−3.733711, 0.50237)	−1.52	0.133
Mean systolic blood pressure (mm/Hg)	−0.00297	(−0.018, 0.01207)	−0.39	0.695
Mean diastolic blood pressure (mm/Hg)	0.0015	(−0.0153, 0.0184)	0.19	0.852
Hba1c (%)	−0.0331	(−0.141, 0.0747)	−0.61	0.543
Average Hba1c%	−0.0343	(−0.1522, 0.0836)	−0.58	0.564
Number of daily glucose measurements	−0.0254	(−0.06926, 0.01835)	−1.16	0.251
Number of weekly hypoglycemic episodes	0.0108	(−0.01295, 0.0346)	0.91	0.368
Insulin (units/kg/day)	0.1097	(−0.553, 0.773)	0.33	0.743
T1D duration (years)	0.0321	(−0.007, 0.0712)	1.63	0.106
Insulin regime **	−0.1098	(−0.4431, 0.223)	−0.66	0.513
Family situation *	−0.2785	(−0.4965, −0.0605)	−2.54	0.013
Laboratory parameters				
Hemoglobin (g/dl)	0.0225	(−0.1173, 0.1624)	0.32	0.749
White blood cells (/mm^3^)	−0.00005	(−0.00017, 0.000057)	−1.01	0.315
Neutrophils percentage (%)	0.0026	(−0.0138, 0.0191)	0.32	0.750
Leukocytes percentage (%)	0.00439	(−0.0134, 0.0222)	0.49	0.625
Platelets (/mm^3^)	4.26 × 10^−7^	(−2.00 × 10^−6^, 2.85 × 10^−6^)	0.35	0.727
Ferritin (ng/mL)	−0.0123	(−0.0234, −0.0013)	−2.50	0.031
Urea (mg/dl)	0.001	(−0.0174, 0.0194)	0.11	0.913
Creatinine (mg/dl)	0.80904	(−0.1425, 1.760619)	1.70	0.094
Triglycerides (mg/dl)	0.00121	(−0.0026, 0.0051)	0.63	0.533
Total cholesterol (mg/dl)	0.0028	(−0.0032, 0.0089)	0.94	0.352
HDL-c (mg/dl)	0.00795	(−0.00412, 0.02002	1.31	0.193
LDL-c (mg/dl)	−0.0007	(−0.0081, 0.0066)	−0.19	0.849
Lp(a) (mg/dl)	−0.0054	(−0.0098, −0.001)	−2.50	0.016
TSH (mIU/L)	0.03588	(−0.04503, 0.11679)	0.88	0.380
Squared leptin	−0.1036	(−0.2022, −0.005)	−2.09	0.040
Log- adiponectin	−0.03921	(−0.27506, 0.1966)	−0.33	0.742
Log-hs-CRP	0.05243	(−0.07789, 0.18275)	0.80	0.426
Inverted-IL6	−0.30806	(−2.5975, 1.9814)	−0.27	0.789

* Lives with both parents, only with mother or with father, with grandparents, with others, or by himself/herself. ** Basal/bolus vs. pump regime. Abbreviations—(95% C.I.): 95% confidence interval; BMI: body mass index; T1D: type-1 diabetes; HbA1c: hemoglobin A1c; TSH: thyroid stimulating hormone; HDL-C: high density lipoprotein-cholesterol; LDL-C: low density lipoprotein-cholesterol; and Lp(a): lipoprotein (a). black: Data are presented as the variable of interest, beta (95% C.I. of beta), T-statistic, and *p*-value.

**Table 2 children-09-01840-t002:** Basic demographic, clinical and biochemical parameters of the study population according to the extreme exercise levels.

(a). Basic Demographic and Clinical Parameters of the Study Population According to the Extreme Exercise Levels			
	Exercise Levels		
Variables [n (%)]	1. Low (n = 20) Mean ± SD Median (Range)	2. High (n = 20) Mean ± SD Median (Range)	*p*-Value
Gender (male/female)	8/12	10/10	0.37 *
Age (years)	15.13 ± 3.41, 15.25 (13.4, 17.8)	15.6 ± 3.63, 16.16 (12.3, 18.4)	0.67 **
Tanner pubertal stage (2/3/4/5)	2/2/1/15	4/1/0/15	0.73 *
Height (cm)	162 ± 12.4, 164 (159, 169.5)	162.5 ± 14.9, 167 (152.5, 174.5)	0.90 **
Weight (kg)	57.2 ± 16.5, 55.5 (48.3, 71.6)	57.5 ± 15.1, 62.1 (44.2, 69.1)	0.95 **
Waist circumference (cm)	74.8 ± 11.3, 73.5 (65.5, 82)	74.5 ± 9.7, 75.5 (66.2, 82)	0.91 **
Hip circumference (cm)	50.3 ± 7.03, 50 (45, 55)	53.9 ± 13.01, 52 (47, 54.5)	0.28 **
Waist-to-hip ratio	1.49 ± 0.17, 1.46 (1.38, 1.64)	1.41 ± 0.19, 1.49 (1.27, 1.56)	0.17 **
Waist-to-height ratio	0.46 ± 0.05, 0.46 (0.43, 0.49)	0.45 ± 0.05, 0.44 (0.42, 0.48)	0.89 **
BMI (kg/m^2^)	21.3 ± 4.0, 20.8 (18.2, 23.2)	21.3 ± 3.02, 21.7 (18.9, 23.8)	0.96 **
SDS-BMI	0.41 ± 0.93, 0.37 (−0.1, 0.99)	0.42 ± 0.93, 0.37 (−0.14, 1.04)	0.98 **
Fat mass percentage	22.5 ± 5.8, 22.2 (18.1, 26.8)	18.9 ± 7.4, 18.4 (15.4, 26.3)	0.10 **
Muscle mass (kg)	43.9 ± 11.9, 41.7 (37.4, 51.7)	46.6 ± 13.1, 48.6 (34.9, 56.7)	0.51 **
Systolic blood pressure (mmHg)	115 ± 9, 114 (109, 120)	113 ± 12, 110 (105, 121)	0.70 **
Diastolic blood pressure (mmHg)	67 ± 6, 69 (63, 72)	68 ± 9, 68 (58, 76)	0.82 **
Mean HbA1c (%)	8.3 ± 2.03, 7.6 (6.9, 9.3)	8.05 ± 1.1, 7.7 (7.1, 9.05)	0.62 **
Daily glucose measurements	6.3 ± 2.5, 5.5 (5, 8.25)	4.9 ± 1.4, 5 (4, 7)	**0.047 ****
Weekly hypoglycemic episodes	2.6 ± 1.9, 2 (1.5, 4)	4.5 ± 11.7, 2 (1.2, 2.7)	0.47 **
Insulin units/kg/day	0.78 ± 0.23, 0.8 (0.64, 0.89)	0.81 ± 0.15, 0.8(0.7, 0.84)	0.68 **
Diabetes duration (years)	4.8 ± 2.6, 5.3 (2.2, 6.8)	6.9 ± 4.3, 6.08 (3.04, 8.62)	0.078 **
Insulin scheme (conventional/pump)	13/7	15/5	0.50 *
Family situation (0/1/2/3)	13/4/2/1	17/3/0/0	0.06 *
**(b). Biochemical Parameters of the Study Population According to the Extreme Exercise Levels**			
	**Exercise Levels**		
**Variables [n (%)]**	**1. Low (n = 20)** **Mean ± SD** **Median (Range)**	**2. High (n = 20)** **Mean ± SD** **Median (Range)**	***p*-Value**
Hemoglobin (g/dl)	13.9 ± 1.2, 13.9 (12.9, 14.5)	14 ± 1.06, 14.2 (13, 14.8)	0.73 **
White blood cells (/mm^3^)	6412 ± 1467, 6440 (5250, 6900)	5954 ± 1451, 5970 (5150, 6400)	0.32 **
Neutrophils percentage (%)	45.7 ± 13.6, 47 (40.2, 56)	46.8 ± 8.9, 47.3 (39.7, 51.4)	0.76 **
Leucocyte percentage (%)	38.9 ± 8.3, 40.5 (33.5, 46.5)	40.4 ± 9.4, 40 (34, 49)	0.66 **
Platelets (X1000)	245 ± 83, 255 (200, 287)	253 ± 78, 257 (209, 290)	0.76 **
Ferritin (ng/mL)	104 ± 17, 104 (92, 117)	31 ± 18, 35 (11, 48)	**0.022 ****
Urea (mg/dl)	27.7 ± 8.4, 26 (20, 32)	28 ± 10.4, 29.3 (22, 36)	0.92 **
Creatinine (mg/dl)	0.61 ± 0.17, 0.66 (0.49, 0.7)	0.71 ± 0.16, 0.71 (0.64, 0.8)	0.084 **
Triglycerides (mg/dl)	62 ± 21, 66 (45, 74)	72 ± 68, 51(42, 78)	0.9 **
Total cholesterol (mg/dl)	149 ± 31, 144 (130, 169)	158 ± 30, 159 (146, 174)	0.41 **
HDL-C (mg/dl)	60 ± 15, 57 (48, 72)	65 ± 15, 63 (52, 78)	0.26 **
LDL-C (mg/dl)	86 ± 27, 81 (69, 105)	85 ± 20, 84 (72, 93)	0.84 **
Lp(a) (mg/dl)	51 ± 66, 16.2 (13, 79)	10 ± 7.4, 7 (5.8, 13.5)	
Log [Lp(a)]	3.23 ± 1.21, 2.78 (2.56, 4.36)	2.07 ± 0.75, 1.94 (1.75, 2.6)	**0.007 ****
TSH (mIU/L)	3.17 ± 1.91, 2.58 (1.8, 3.86)	3.73 ± 3.07, 3.08 (2.09, 3.95)	0.48 **
Leptin (ng/mL)	13.51 ± 13.09, 7.9 (3.8, 19.7)	7.52 ± 8.92, 3.97 (0.69, 12.7)	
SqR(Leptin)	3.26 ± 1.72, 2.8 (1.95, 4.44)	2.24 ± 1.62, 1.98 (0.83, 3.57)	**0.059 ****
Adiponectin (ng/mL)	31,223 ± 22,808, 22,647 (12,660, 43,664)	26,858 ± 18,405, 18,758 (14,165, 34,860)	
Log (Adiponectin)	10.08 ± 0.76, 10.02 (9.4, 10.67)	10.01 ± 0.58, 9.83 (9.55, 10.45)	0.74 *
hs-CRP (mg/dl)	1653 ± 2635, 993 (300, 1799)	2101 ± 3382, 1107 (732, 1738)	
Log(hs-CRP)	6.61 ± 1.34, 6.9 (5.7, 7.48)	6.92 ± 1.25, 7.0 (6.59, 7.46)	0.45 **
IL-6 (pg/mL)	11.4 ± 15.1, 6.6 (5.2, 9.6)	27.9 ± 59.5, 6.3 (4.9, 13.5)	
1/IL-6	0.14 ± 0.06, 0.15 (0.1, 0.19)	0.13 ± 0.08, 0.15 (0.07, 0.2)	0.79 **

* Fisher’s exact test and ** ANOVA with Bonferroni correction. Abbreviations—(95% C.I.): 95% confidence interval; BMI: body mass index; T1D: type-1 diabetes; HbA1c: hemoglobin A1c; TSH: thyroid stimulating hormone; HDL-C: high density lipoprotein-cholesterol; LDL-C: low density lipoprotein-cholesterol; and Lp(a): lipoprotein (a). black: Data are presented as the variable of interest, beta (95% C.I. of beta), T-statistic, and *p*-value. Family situation: lives with both parents, only with mother or with father, with grandparents, with others, or by himself/herself.

## Data Availability

The authors confirm that the majority of the data supporting the findings of this study are available within the article. Raw data are available from the authors upon reasonable request.

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
