# Peer review of "The Role of Exercise on Cardiometabolic Profile and Body Composition in Youth with Type 1 Diabetes"

_children, 2022, doi:10.3390/children9121840_

Round 1

Reviewer 1 Report

This study evaluated the effect of physical activity on cardiometabolic profile, inflammatory cytokines, adipokines, and also infection indices in children and youth with Type 1 diabetes. 

The topic is interesting, although suffers structural and perhaps conceptual defects.

Please find below my comments:

Line 28: „Family situation” it is unclear from the abstract whether this wording refers to the socioeconomic situation, legal situation, or other?

Line 28: Lp(a) - this abbreviation should be explained

Line 58: HDL - this abbreviation should be explained

Line 62: LDL-c - this abbreviation should be explained

Line 140-141: „No ethical concerns were risen due to the observational nature of this study” 

All research involving human participants requires ethical approval by the Ethics Committees. In MDPI Children Journal ethical approval is not applicable only for studies not involving humans or animals. In the present research, the study group is humans, and the biological material (blood) is analyzed, therefore ethical approval is required.

Line 158: Results presentation is very poor.

- The tables should be located directly at the place of their description. Not at the end of the results section

- The descriptions should be under the figures

Line 159: There are duplicated results from the previous publication. This fact should be clearly highlighted in the text

Lines: 46 and 207: The same information, and quoted different sources. Please verify.

Moreover:

- The study was conducted for only a week.

- I am not convinced if the body composition analyzer Tanita BC-418 MA is intended for people under 18 years of age. According to manufacturer information about the device: „Applications: This equipment can be used in the screening of certain adult diseases and conditions related to body weight and composition”. 

- There is a significant age dispersion in the study group. It is difficult to compare the results obtained from a 6 -year -old child to the results of a 21 -year -old person.

- The results of the body composition analysis have not been described in the manuscript

- The discussion section only presents articles confirming the obtained results. Most often it was only one article. There is a lack of counterarguments.

Author Response

Dear Reviewer,

We would like to thank you for your valuable and important comments, that have resulted in the significant improvement of our paper “The role of exercise on cardiometabolic profile and body com-position in youth with Type 1 Diabetes”. The answers to the questions are included below each question.

With grateful thanks for your time and kind consideration,

Yours Sincerely,

Prof. Dr. Kyriaki Karavanaki

Professor in Pediatrics, Pediatric Endocrinology and Diabetes,

Head of Diabetes and Metabolism Clinic,

2nd Department of Pediatrics, National and Kapodistrian University of Athens,

“P. & A. Kyriakou” Children’s Hospital of Athens, Greece.

RESPONSES:

Line 28: „Family situation” it is unclear from the abstract whether this wording refers to the socioeconomic situation, legal situation, or other?

Response: Thank you very much for your comment. The abstract has been amended. Furthermore, family situation is explained both in results and in the tables (Line 174-175).

Line 28: Lp(a) - this abbreviation should be explained

Response: Thank you very much for your comment.  The abbreviations spell out has been amended in the abstract and in the text.

Line 58: HDL - this abbreviation should be explained

Response: Thank you very much for your comment.  The abbreviations spell out has been amended.

Line 62: LDL-c - this abbreviation should be explained

Response: Thank you very much for your comment.  The abbreviations spell out has been amended.

Line 140-141: „No ethical concerns were risen due to the observational nature of this study”

All research involving human participants requires ethical approval by the Ethics Committees. In MDPI Children Journal ethical approval is not applicable only for studies not involving humans or animals. In the present research, the study group is humans, and the biological material (blood) is analyzed, therefore ethical approval is required.

Response: Thank you very much for your comment. Ethical approval was obtained and was included in Institutional Review Board Statement (Line 349-350). Nevertheless, Ethics was amended accordingly and the approval from Ethics Committee was added also in the Ethics Section.

Institutional Review Board Statement: The study was approved by the Ethics Committee of the “P&A Kyriakou” Children’s Hospital, Athens, Greece, protocol number: 12919, date of approval: 24 July 2019.

Line 158:

The tables should be located directly at the place of their description. Not at the end of the results section. The descriptions should be under the figures

Response: Thank you very much for your comment.  The results have been amended accordingly.

Line 159: There are duplicated results from the previous publication. This fact should be clearly highlighted in the text

Response: Thank you very much for your comment.  The demographic parameters have been used in our previous publication in “Children” journal and reference to our previous paper is added accordingly.

Lines: 46 and 207: The same information and quoted different sources. Please verify.

Response: Thank you very much for your comment.  The references have been amended accordingly.

The study was conducted for only a week.

Response: Thank you very much for your comment.  Many studies including pedometers in children and adolescents are conducted in a week as the use of such devices in children are difficult to be used for longer periods by the participants (difficult for children and adolescents to comply for longer periods), furthermore most pedometers have a seven day memory card [1–4].

- I am not convinced if the body composition analyzer Tanita BC-418 MA is intended for people under 18 years of age. According to manufacturer information about the device: „Applications: This equipment can be used in the screening of certain adult diseases and conditions related to body weight and composition”.

Response: Thank you very much for your comment.

I have contacted “Tanita BC-418 MA” advisors and they kindly provided a few research articles of Tanita BC-418 MA used in paediatric population [5–8]. Furthermore, Tanita BC-418 MA has been safely used for the past years in Department of Endocrinology Growth and Development, “P&A Kyriakou” Children’s Hospital, Athens, Greece, which is a tertiary level children’s Hospital in Greece.

- There is a significant age dispersion in the study group. It is difficult to compare the results obtained from a 6 -year -old child to the results of a 21 -year -old person.

Response: Thank you very much for your comment. Due to the age dispersion of the paediatric T1D clinics, most studies regarding T1D children and adolescents have a similar age dispersion to our study [9–13].

- The results of the body composition analysis have not been described in the manuscript

Response: Thank you very much for your comment.  The results of fat mass and muscle mass are included in the results section and the tables.

- The discussion section only presents articles confirming the obtained results. Most often it was only one article. There is a lack of counterarguments.

Response: Thank you very much for your comment. Unfortunately there is a raff limit of references from “Children Journal” which is around 50 per article and unfortunately we have already excided the limit by 15 references (we totally included 65 references).

References:

  1. Sundberg, F.; Forsander, G.; Fasth, A.; Ekelund, U. Children Younger than 7 Years with Type 1 Diabetes Are Less Physically Active than Healthy Controls: Physical Activity in Young Children with Type 1 Diabetes. Acta Paediatr. 2012, 101, 1164–1169, doi:10.1111/j.1651-2227.2012.02803.x.
  2. Baquet, G.; Stratton, G.; Van Praagh, E.; Berthoin, S. Improving Physical Activity Assessment in Prepubertal Children with High-Frequency Accelerometry Monitoring: A Methodological Issue. Prev. Med. 2007, 44, 143–147, doi:10.1016/j.ypmed.2006.10.004.
  3. Namburar, S.; Checkley, W.; Flores-Flores, O.; Romero, K.M.; Fraser, K.T.; Hansel, N.N.; Pollard, S.L.; __ Risk Factors for Physical Inactivity Among Children With and Without Asthma Living in Peri-Urban Communities of Lima, Peru. J. Phys. Act. Health 2020, 17, 816–822, doi:10.1123/jpah.2019-0553.
  4. Sigmundová, D.; Sigmund, E.; Badura, P.; Hollein, T. Parent-Child Physical Activity Association in Families with 4- to 16-Year-Old Children. Int. J. Environ. Res. Public. Health 2020, 17, 4015, doi:10.3390/ijerph17114015.
  5. Prins, M.; Hawkesworth, S.; Wright, A.; Fulford, A.J.C.; Jarjou, L.M.A.; Prentice, A.M.; Moore, S.E. Use of Bioelectrical Impedance Analysis to Assess Body Composition in Rural Gambian Children. Eur. J. Clin. Nutr. 2008, 62, 1065–1074, doi:10.1038/sj.ejcn.1602830.
  6. Ramírez-Vélez, R.; Correa-Bautista, J.; Martínez-Torres, J.; González-Ruíz, K.; González-Jiménez, E.; Schmidt-RioValle, J.; Garcia-Hermoso, A. Performance of Two Bioelectrical Impedance Analyses in the Diagnosis of Overweight and Obesity in Children and Adolescents: The FUPRECOL Study. Nutrients 2016, 8, 575, doi:10.3390/nu8100575.
  7. McCarthy, H.D.; Samani-Radia, D.; Jebb, S.A.; Prentice, A.M. Skeletal Muscle Mass Reference Curves for Children and Adolescents. Pediatr. Obes. 2014, 9, 249–259, doi:10.1111/j.2047-6310.2013.00168.x.
  8. Devakumar, D.; Grijalva-Eternod, C.S.; Roberts, S.; Chaube, S.S.; Saville, N.M.; Manandhar, D.S.; Costello, A.; Osrin, D.; Wells, J.C.K. Body Composition in Nepalese Children Using Isotope Dilution: The Production of Ethnic-Specific Calibration Equations and an Exploration of Methodological Issues. PeerJ 2015, 3, e785, doi:10.7717/peerj.785.
  9. Wilkie, L.; Mitchell, F.; Robertson, K.; Kirk, A. Motivations for Physical Activity in Youth with Type 1 Diabetes Participating in the ActivPals Project: A Qualitative Study. Pract. Diabetes 2017, 34, 151–155, doi:10.1002/pdi.2107.
  10. Tsalikian, E.; Mauras, N.; Beck, R.W.; Tamborlane, W.V.; Janz, K.F.; Chase, H.P.; Wysocki, T.; Weinzimer, S.A.; Buckingham, B.A.; Kollman, C.; et al. Impact of Exercise on Overnight Glycemic Control in Children with Type 1 Diabetes Mellitus. J. Pediatr. 2005, 147, 528–534, doi:10.1016/j.jpeds.2005.04.065.
  11. The Diabetes Research in Children Network (DirecNet) Study Group The Effects of Aerobic Exercise on Glucose and Counterregulatory Hormone Concentrations in Children With Type 1 Diabetes. Diabetes Care 2006, 29, 20–25, doi:10.2337/diacare.29.01.06.dc05-1192.
  12. Varni, J.W.; Burwinkle, T.M.; Jacobs, J.R.; Gottschalk, M.; Kaufman, F.; Jones, K.L. The PedsQLTM in Type 1 and Type 2 Diabetes: Reliability and Validity of the Pediatric Quality of Life InventoryTM Generic Core Scales and Type 1 Diabetes Module. Diabetes Care 2003, 26, 631–637, doi:10.2337/diacare.26.3.631.
  13. Aljawarneh, Y.M.; Wardell, D.W.; Wood, G.L.; Rozmus, C.L. A Systematic Review of Physical Activity and Exercise on Physiological and Biochemical Outcomes in Children and Adolescents With Type 1 Diabetes. J. Nurs. Scholarsh. 2019, 51, 337–345, doi:10.1111/jnu.12472.

Reviewer 2 Report

Larger sample size will be useful in future studies.

Author Response

Thank you.

Reviewer 3 Report

Study well done, although no control group of non-diabetic was good to have in the study, still the outcome of the study is important based on the design of the study.

Author Response

Thank you.

Round 2

Reviewer 1 Report

Much thanks to the authors for incorporating major recommendations from the review, with the paper better suitable for publication.

I have two added further comments of content:

1) Why the authors did not include approval from the Ethics Committee in the previous version of the manusctipt? Could you attach in the system scan of the protocol of the Ethics Committee decision with the research title?

2) Newly added information on body composition analysis should be also described in the results section.

Author Response

RESPONSE TO REVIEWER:

Dear Reviewer,

We would like to thank you for your valuable and important comments, that have resulted in the significant improvement of our paper “The role of exercise on cardiometabolic profile and body com-position in youth with Type 1 Diabetes”. The answers to the questions are included below each question.

With grateful thanks for your time and kind consideration,

Yours Sincerely,

Prof. Dr. Kyriaki Karavanaki

Professor in Pediatrics and Pediatric Diabetes, Diabetes and Metabolism Clinic,

2nd Department of Pediatrics, National and Kapodistrian University of Athens,

“P. & A. Kyriakou” Children’s Hospital of Athens, Greece.

RESPONSES:

1) Why the authors not include approval from the Ethics Committee in the previous version of the manusctipt?

Could you attach in the system scan of the protocol of the Ethics Committee decision with the research title?

Response: Thank you very much for your comment. Ethical approval was obtained and was included in Institutional Review Board Statement (Line 349-350) of the previous version of manuscript. Nevertheless, the approval from Ethics Committee was added also in the Ethics Section. Furthermore, we have attached the Ethics Committee decision in the online system.

Institutional Review Board Statement: The study was approved by the Ethics Committee of the “P&A Kyriakou” Children’s Hospital, Athens, Greece, protocol number: 12919, date of approval: 24 July 2019.

2) Newly added information on body composition analysis should be also described in the results section.

Response: Thank you very much for your comment.  The results of fat mass and muscle mass are now included in the results section and the tables.
